# Analysis of horizontal dynamic response of single pile under vertical load-Rayleigh wave combined action

Sitong Zhao[1,2*], Xiaohui Zhang[3], Hui Yan[4], Ruming Ma[5*], Fanbao Meng[6], Zhiqiang Wang[7]

1 Gansu Changlong Highway Maintenance Technology Institute Co., Ltd., Lanzhou, Gansu, China, 2 School of civil engineering, Lanzhou University of Technology, Lanzhou, Gansu, China, 3 The Second Construction Company Ltd. of China Construction Second Bureau, Shenzhen, China, 4 China Construction Fifth Engineering Bureau Co.Ltd, Changsha, China, 5 The First Exploration Team of Shandong Coalfield Geologic Bureau,Qingdao,China, 6 Shandong Engineering Research Center for the Prevention and Control of Mine Gas Disasters,Qingdao,China, 7 China National Building Materials (Gansu) Survey, Planning and Design Co., LTD, Tianshui, Gansu

* 15002599635@163.com (SZ); 317993801@qq.com (RM)

## Abstract

In practical engineering, pile foundation will be subject to the combined action of vertical and Rayleigh wave, and the existence of vertical dynamic load will cause the second-order effect that will lead to the increase of horizontal displacement. This paper presents a computational model for the effect of vertical load on the lateral response of monopole under Rayleigh wave. The pile cap is equivalent to a rigid block, and the constraint of pile head is regarded as a flexible constraint. By means of operator decomposition theory and variable separation method, the horizontal dynamic response of single-phase soil in uniform free field under Rayleigh wave propagation is obtained. The closed form solution of soil resistance under the combined action of Rayleigh wave and vertical load is obtained by means of operator decomposition theory and variable separation method. Based on Timoshenko beam theory, the dynamic differential equation of pile considering vertical load is established. A comparison with an existing solution is performed to verify the proposed solution. Through numerical examples, the effects of the vertical load, flexible constraints, dimensionless frequency and Poisson's ratio on the lateral response of monopile are assessed.

## 1 Introduction

Rayleigh waves propagate along the surface of the earth, which is the most harmful and fast in natural earthquake [1–2]. As a surface wave, Rayleigh waves are more likely to be generated in steady state harmonic loads and transient excitation under construction sites, highway traffic and machine operation conditions [3–4]. To study

**Data availability statement:** The data supporting the conclusions of this study are derived from the formula derivations presented in the main text. During the research process, the relevant variables and parameters involved in the formula derivations were subjected to specific numerical calculations and processing using MATLAB software. All the final data obtained through these derivations and calculations are compiled in the supplementary attachment named "data.xls". The data in the file are organized with clear labels and instructions, ensuring their accessibility and usability for interested researchers.

**Funding:** This work was supported by the National Natural Science Foundation of China (.52408368). Lanzhou University of Technology Hongliu outstanding young talent support program, Key R&D Plan of Gansu Province(23YFFA0063). 2024 University teacher Innovation Fund project (2024A-019). Gansu Provincial Key R&D Program (25YFGE002). Tianshui Science and Technology Support Program (TS-STK-2024A-283).

the characteristics of Rayleigh wave propagation, a large number of scholars have carried out theoretical analysis [5–7], numerical model [8–10]and experimental research [11–12].

To clarify the propagation characteristics and attenuation laws of Rayleigh waves, scholars first studied the propagation of Rayleigh waves in soil, including single-phase soil [13], saturated soil [14–15], and unsaturated soil [16]. Based on the interaction between tunnel and soil, Zhao et al. studied the dynamic response of tunnel under the action of Rayleigh wave, and gave a closed-form solution [17]. Due to the great harm of Rayleigh wave, a large number of scholars have proposed a variety of methods for vibration isolation of Rayleigh wave, including open trench [18–19]. The research on soil-structure interaction focuses on the dynamic coupling of FGM structures, elastic foundations and moving loads [20–22]. Gao et al. [23] studied the vibration isolation effect of WIB in saturated soil and analyzed the interaction between saturated soil-ground-wave-baffle. Ma et al. [24] studied the vibration isolation effect of saturated porous gradient WIB with liquid in saturated soil foundation under moving load. On the basis of the previous research, Zhou et al. [25] proposed a new vibration isolation method based on the combination of open trench and wave impeding block.

Pile foundation is widely used in high-rise buildings, bridges, ports and offshore platforms due to its high bearing capacity, the ability to control deformation effectively and improve seismic performance of buildings [26–29]. It is inevitable to study the effect of Rayleigh waves on piles due to the great harm of Rayleigh waves. Yang et al. [30] considering the flexible confinement of pile top, the dynamic response of unsaturated soil-pile system under the action of Rayleigh wave is studied. Zhang et al. [31] studied the dynamic response of piles in the deep-sea environment under the action of Rayleigh waves, considering that the offshore foundation is a two-phase saturated soil medium. M Bahrami et al. [32] studied the dynamic response of Rayleigh waves acting on monopiles in shallow underground by Fourier transform. Liu et al. [33] studied the effect of Rayleigh wave propagation on the longitudinal vibration characteristics of pipe piles in saturated soil under non-isothermal conditions. In addition, considering that the pile head is a flexible constraint. Cui et al. [34–35] established an analytical solution for the horizontal vibration of floating piles in radially heterogeneous saturated soil using the Biot dynamic equation. The analytical solution of the horizontal impedance of the pile head was obtained through the Laplace method and the separation of variables method, combined with the continuity of soil displacement and stress equilibrium conditions.

From the above discussion, it can be found that: due to the complexity of analyzing the horizontal dynamic response of piles under Rayleigh waves considering the effect of vertical loads, the influence of vertical loads is ignored in the theoretical calculation model. Consequently, this study established a calculation model considering the influence of vertical loads on the horizontal dynamic response of piles under Rayleigh waves. The results of this study provide a more practical technical method for the dynamic analysis and design of pile foundations under the action of Rayleigh waves.

## 2 Computational model

### 2.1 Description of the problem

The mathematical model of soil-pile system under the action of Rayleigh wave is represented by column coordinates, as shown in Fig 1, in which the pile length is $H$, radius is $r_0$, cross-sectional area is $A_p$, density is $\rho_p$, Young's modulus is $E_p$ and shear modulus is $G_p$. The pile is modeled as a Timoshenko beam with a fixed end boundary condition at the pile end, and the pile top is regarded as a rigid block with a mass of $M$. The pile surrounding soil is regarded as a linear elastic material, and Poisson's ratio, damping ratio and Shear modulus are $v_s$, $\xi$ and $G_s$, respectively. Rayleigh wave is transmitted to the pile through the soil in the form of waves under the action of Rayleigh wave.

To study the influence of Rayleigh waves on the horizontal dynamic response of pile foundation under the flexible confinement state of pile top, the constraint degree of pile top is expressed by rotational stiffness:

$$M_P(0) = k_0\theta_c + k_1(\theta_p(0) - \theta_c) \tag{1}$$

where, $k_0$ and $k_1$ represent the initial stiffness and strengthened stiffness, respectively. $\theta_p$ and $\theta_c$ are the rotation angles corresponding to the pile head bending moment and the yield bending moment, respectively.

### 2.2 Basic equations

The Cauchy stress equation indicates that the motion equation are:

$$\sigma_{ij,j} = \rho_s\ddot{u}_i \tag{2}$$

where $\rho_s$ is density of soil, $\sigma_{ij}$ are the stress tensor of the soil, $u_i$ is the displacement component of soil.

In cylindrical coordinates, the geometric relations are:

$$\varepsilon_{ij} = \frac{1}{2}(u_{i,j} + u_{j,i}) \tag{3}$$

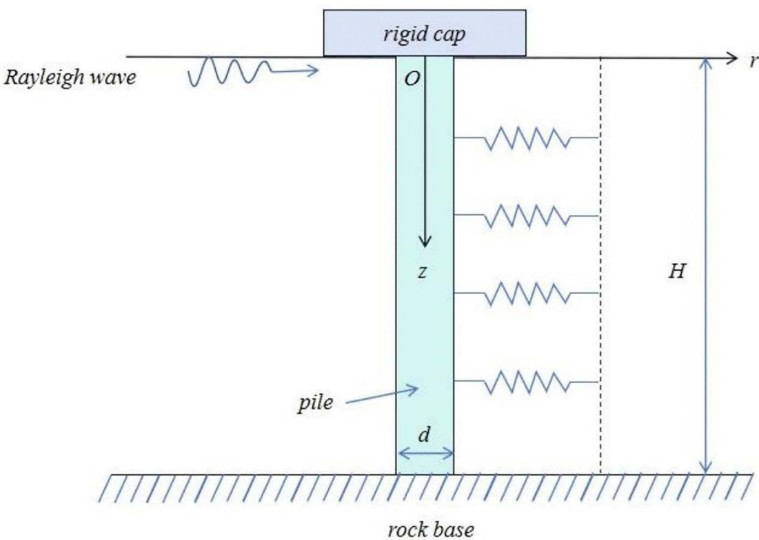

**Fig 1. Pile-soil interaction calculation model.**

For isotropic medium, the physical equations for single-phase soils are:

$$\sigma_{ij} = 2\mu\varepsilon_{ij} + \lambda\varepsilon_{kk}\delta_{ij} \tag{4}$$

where $u_{ij}$ are the displacement tensor of the soil, $\varepsilon_{ij}$ are the strain tensor of the soil; $\lambda$ and $\mu$ are Lame constants, respectively. $\delta_{ij}$ represents the Kronecker function

Based on the Timoshenko beam theory, by taking the horizontal force balance of the pile section, the control differential equation of the pile section is established as follows [30]:

$$A_p G_p k' \left( \frac{\partial^2 u_p}{\partial z^2} - \frac{\partial \theta_p}{\partial z} \right) + \rho_p A_p \frac{\partial^2 u_p}{\partial t^2} + (k_h + ic_h)(u_p - u_r \cos\theta) = 0 \tag{5}$$

$$A_p G_p k' \left( \frac{\partial u_p}{\partial z} - \theta_p \right) - \rho_p I_p \frac{\partial^2 u_p}{\partial t^2} - E_p I_p \frac{\partial^2 \theta_p}{\partial z^2} + A_p P \frac{\partial u_p}{\partial z} = 0 \tag{6}$$

where, $u_p(z,t) = \bar{u}_p(z)e^{i\omega t}$ and $\theta_p(z,t) = \bar{\theta}_p(z)e^{i\omega t}$ represent the horizontal displacement and rotation angle of the pile, respectively; $k'$ denotes the modified shear factor in Timoshenko beam theory, $P$ is the vertical load caused by the mass of the superstructure. $k_h$ and $c_h$ are the stiffness and damping of soil on piles under the action of Rayleigh waves.

### 2.3 Boundary conditions

The surface of the soil layer is a free soil layer, so the stress at $z = 0$ are zero:

$$\sigma_z\big|_{z=0} = 0, \quad \tau_{rz}\big|_{z=0} = 0 \tag{7}$$

The bottom of the soil layer is fixed, so the radial and tangential displacement of the soil at $z = H$ are zero:

$$u_r\big|_{z=H} = 0, \quad u_\theta\big|_{z=H} = 0 \tag{8}$$

At the pile-soil interface($r = r_0$), the radial and tangential displacement components can be expressed as:

$$u_r\big|_{r=r_0} = u_p \cos\theta, \quad u_\theta\big|_{r=r_0} = -u_p \sin\theta \tag{9}$$

where $u_p$ is the lateral displacement of the pile.

### 3 Model development

According to the calculation method provided by Cao et al. [38], the following expression can be obtained.

$$\varepsilon_z = -\frac{\nu}{1-\nu}(\varepsilon_r + \varepsilon_\theta) \tag{10}$$

Substituting equation (10) into (4) yields:

$$\sigma_r = \mu\left[\eta_a^2(\varepsilon_r + \varepsilon_\theta) - 2\varepsilon_\theta\right] \tag{11}$$

$$\sigma_\theta = \mu\left[\eta_a^2(\varepsilon_r + \varepsilon_\theta) - 2\varepsilon_r\right] \tag{12}$$

where, $\eta_a^2 = 2/(1-\nu)$.

Substituting equations (3), (4), (10)-(12) into (2) yields:

$$\eta_b^2 \frac{\partial}{\partial r}\left[\frac{1}{r}\left(\frac{\partial(ru_r)}{\partial r} + \frac{\partial u_\theta}{\partial \theta}\right)\right] - \frac{1}{r^2}\frac{\partial}{\partial \theta}\left[\frac{\partial(ru_\theta)}{\partial r} - \frac{\partial u_r}{\partial \theta}\right] + \frac{\partial^2 u_r}{\partial z^2} + \frac{\rho_s \omega^2}{G_s}u_r = 0 \tag{13}$$

$$\frac{\partial}{\partial r}\left[\frac{1}{r}\left(\frac{\partial(ru_\theta)}{\partial r} - \frac{\partial u_r}{\partial \theta}\right)\right] + \eta_b^2\frac{1}{r^2}\frac{\partial}{\partial \theta}\left[\frac{\partial(ru_r)}{\partial r} + \frac{\partial u_\theta}{\partial \theta}\right] + \frac{\partial^2 u_\theta}{\partial z^2} + \frac{\rho_s \omega^2}{G_s}u_\theta = 0 \tag{14}$$

where, $\eta_b^2 = (2-\nu)/(1-\nu)$.

The horizontal displacement and tangential displacement are expressed by two potential functions ++$\phi$ and $\psi$:

$$u_r = \frac{\partial \phi}{\partial r} + \frac{1}{r}\frac{\partial \psi}{\partial \theta}$$

$$u_\theta = \frac{1}{r}\frac{\partial \phi}{\partial \theta} - \frac{\partial \psi}{\partial r} \tag{15}$$

Substituting equation (15) into equation (13) and (14) yields:

$$\eta_b^2\frac{\partial}{\partial r}\left(\nabla^2\phi\right) + \frac{1}{r}\frac{\partial}{\partial \theta}\left(\nabla^2\psi\right) + \frac{\partial^2}{\partial z^2}\left(\frac{\partial \phi}{\partial r} + \frac{1}{r}\frac{\partial \psi}{\partial \theta}\right) + \frac{\rho_s}{G_s}\omega^2\left(\frac{\partial \phi}{\partial r} + \frac{1}{r}\frac{\partial \psi}{\partial \theta}\right) = 0 \tag{16}$$

$$-\frac{\partial}{\partial r}\left(\nabla^2\psi\right) + \eta_b^2\frac{1}{r}\frac{\partial}{\partial \theta}\left(\nabla^2\phi\right) + \frac{\partial^2}{\partial z^2}\left(\frac{1}{r}\frac{\partial \phi}{\partial \theta} - \frac{\partial \psi}{\partial r}\right) + \frac{\rho_s}{G_s}\omega^2\left(\frac{1}{r}\frac{\partial \phi}{\partial \theta} - \frac{\partial \psi}{\partial r}\right) = 0 \tag{17}$$

Combined formula (16), (17) obtained:

$$\eta_b^2\nabla^2\phi + \frac{\partial^2\phi}{\partial z^2} + \frac{\rho_s}{G_s}\omega^2\phi = 0 \tag{18}$$

$$\nabla^2\psi + \frac{\partial^2\psi}{\partial z^2} + \frac{\rho_s}{G_s}\omega^2\psi = 0 \tag{19}$$

Through operator decomposition theory and variable separation method, take $\phi$ and $\psi$ as:

$$\phi = A_s \exp\left(-sz - ik_R r\cos\theta\right) \tag{20}$$

$$\psi = B_s \exp\left(-\gamma z - ik_R r\cos\theta\right) \tag{21}$$

where, $A_s$ and $B_s$ are the undetermined constants related to the boundary conditions; $k_R$, $V_R = \omega/k_R$ denote the complex wave number and phase velocity of Rayleigh waves, respectively; Among these parameters, the complex wave number $k_R$ is determined by solving the characteristic equation for Rayleigh waves, with its complex representation accounting for wave attenuation due to material damping. $s = \sqrt{k_R^2 - k_p^2}$ and $\gamma = \sqrt{k_R^2 - k_s^2}$ are attenuation exponent corresponding to the compression wave and the shear wave, respectively. where $k_p^2 = \frac{1-\nu}{2-\nu}\frac{\rho_s\omega^2}{G_s}$, $k_s^2 = \frac{\rho_s\omega^2}{G_s}$.

The free surface stress boundary conditions for soil is:

$$\tau_{rz}\big|_{z=0} = 0 \tag{22}$$

By substituting the geometric equation (3), the physical equation (4) and the potential function (15) into the equations (20), (21), and finally combining the boundary conditions (22) to obtain:

$$\tau_{rz} = G_s \left[ \begin{array}{l} ik_R \cos\theta s A_s \exp(-sz - ik_R r \cos\theta) - \gamma ik_R \sin\theta B_s \exp(-\gamma z - ik_R r \cos\theta) \\ -\frac{H\nu_s}{1-\nu_s} ik_R^3 \cos\theta A_s \exp(-sz - ik_R r \cos\theta) \end{array} \right] \tag{23}$$

Substituting the boundary condition with the above equation yields:

$$\int_0^{\frac{\pi}{2}} \tau_{rz} d\theta = 0 \tag{24a}$$

$$\left( s - \frac{H\nu_s}{1-\nu_s} k_R^2 \right) A_s - \gamma B_s = 0 \tag{24b}$$

$$B_s = dA_s \tag{24c}$$

where, $d = \left( s - \frac{H\nu_s}{1-\nu_s} k_R^2 \right) \Big/ \gamma$.

Combining equations (15), (20), (21) and (24), the displacement of free-field soil under the action of Rayleigh waves can be expressed as:

$$\begin{array}{l} u_r = -ik_R \cos\theta A_s \exp(-sz - ik_R r \cos\theta) \\ \quad + dik_R \sin\theta A_s \exp(-\gamma z - ik_R r \cos\theta) \end{array} \tag{25}$$

$$\begin{array}{l} u_\theta = ik_R \sin\theta A_s \exp(-sz - ik_R r \cos\theta) \\ \quad + dik_R \cos\theta A_s \exp(-\gamma z - ik_R r \cos\theta) \end{array} \tag{26}$$

where, $t_1 = -ik_R \cos\theta$, $t_2 = dik_R \sin\theta$.

Therefore, the displacement of soil in the free-field under the action of Rayleigh waves is:

$$u_r = A_s \left[ t_1 \exp(-sz - ik_R r \cos\theta) + t_2 \exp(-\gamma z - ik_R r \cos\theta) \right] \tag{27}$$

$$u_\theta = A_s \left[ \frac{t_2}{d} \exp(-sz - ik_R r \cos\theta) - t_1 d \exp(-\gamma z - ik_R r \cos\theta) \right] \tag{28}$$

## 4 Dynamic response of piles under Rayleigh wave

### 4.1 Horizontal resistance

The dynamic movement of the pile under the action of Rayleigh wave is determined by the surrounding soil. Zhang et al. [36], studied the horizontal dynamic response of pile group unsaturated soil, verified the approximate Winkler model

solution of the pile head impedance function in the frequency domain, and derived the pile head impedance function in single-phase soil based on this research. The dynamic Winkler model is used to describe the horizontal dynamic response of the single-phase soil-pile system. The horizontal resistance on the per unit length of pile is [37]:

$$q_h = (k_h + ic_h)\, u_{p0} \tag{29}$$

Continuity conditions for the displacement of pile and soil interface are:

$$u_r = u_{p0}\cos\theta,\ u_\theta = -u_{p0}\sin\theta \tag{30}$$

where, $u_{p0}$ denotes the dimensionless horizontal displacement amplitude of the pile in the direction of $\theta = 0$.

Combining boundary conditions yields:

$$u_r = A_s(-ik_R\cos\theta e^{-sz-ik_Rr\cos\theta} + ik_R\sin\theta de^{-\gamma z-ik_Rr\cos\theta}) = u_{p0}\cos\theta \tag{31}$$

$$u_\theta = A_s(ik_R\sin\theta e^{-sz-ik_Rr\cos\theta} + ik_R\cos\theta de^{-\gamma z-ik_Rr\cos\theta}) = -u_{p0}\sin\theta \tag{32}$$

thereby:

$$
\begin{aligned}
A_s(-ik_Re^{-sz-ik_Rr\cos\theta} + ik_R\tan\theta de^{-\gamma z-ik_Rr\cos\theta}) &= u_{p0}\\
A_s(ik_Re^{-sz-ik_Rr\cos\theta} + ik_R\cot\theta de^{-\gamma z-ik_Rr\cos\theta}) &= -u_{p0}\\
\begin{bmatrix} -ik_Re^{-sz-ik_Rr\cos\theta} & ik_R\tan\theta de^{-\gamma z-ik_Rr\cos\theta}\\ ik_Re^{-sz-ik_Rr\cos\theta} & ik_R\cot\theta de^{-\gamma z-ik_Rr\cos\theta} \end{bmatrix} &= \begin{bmatrix} A_s\\ A_s \end{bmatrix} = \begin{bmatrix} u_{p0}\\ -u_{p0} \end{bmatrix}
\end{aligned}
\tag{33}
$$

solved by the above formula:

$$A_s = ku_{p0},\ \ k = \frac{1}{-ik_Re^{-ik_R0.5}} \tag{34}$$

The lateral resistance per unit length of the pile under the action of harmonic force when horizontal is:

$$q_h = -\int_0^{2\pi}(\sigma_r\cos\theta - \tau_{r\theta}\sin\theta)d\theta \tag{35}$$

combined equations (3), (4), (11) and (27) and (28) are obtained:

$$\sigma_r = \mu\left[\eta_a^2\left(-k_R^2A_se^{-sz-ik_Rr\cos\theta}\right) + 2\left(k_R^2\sin^2\theta A_se^{-sz-ik_Rr\cos\theta} + k_R^2\sin\theta\cos\theta dA_se^{-\gamma z-ik_Rr\cos\theta}\right)\right] \tag{36}$$

$$\tau_{r\theta} = G_s\left[2k_R^2\sin\theta\cos\theta A_se^{-sz-ik_Rr\cos\theta} + \left(\cos^2\theta - \sin^2\theta\right)dk_R^2A_se^{-\gamma z-ik_Rr\cos\theta}\right] \tag{37}$$

Substituting equation (34) into equation (36), (37) and combining equation (35) to solves the horizontal resistance on the per unit length of pile:

$$q_h = -\int_0^{2\pi}(\sigma_r\cos\theta - \tau_{r\theta}\sin\theta)d\theta = (k_h + ic_h)\, u_{p0} \tag{38}$$

where $k_h$ and $c_h$ represent the spring rate and damping coefficient in the dynamic Winkler model, respectively.

## 4.2 The Dynamic response of piles considering the influence of vertical loads

Substituting the horizontal displacement equation (27) and horizontal resistance on the per unit length of pile equation (29) into (5) and (6), and omitting the common time-harmonic factor $e^{i\omega t}$, decoupling (5) and (6) to obtain two independent equations:

$$\frac{d^4\bar{u}_p}{dz^4} + K\frac{d^2\bar{u}_p}{dz^2} + N\bar{u}_p = -\frac{(k_h+ic_h)}{k'A_pG_p}\frac{d^2(u_r\cos\theta)}{dz^2} \\ -\left(\frac{\rho_p\omega^2}{k'A_pG_pE_p} - \frac{1}{E_pI_p}\right)(k_h+ic_h)\,u_r\cos\theta \tag{39}$$

$$\frac{d^4\bar{\theta}_p}{dz^4} + K\frac{d^2\bar{\theta}_p}{dz^2} + N\bar{\theta}_p = -\frac{(k_h+ic_h)(k'G_p+P)}{k'G_pE_pI_p}\frac{d^2(u_r\cos\theta)}{dz} \tag{40}$$

where, $K = \frac{\rho_p\omega^2}{E_p} + \frac{\rho_p\omega^2}{k'G_p} + \frac{A_pP}{E_pI_p} - \frac{(k_h+ic_h)}{k'A_pG_p}$, $N = \frac{\rho_p^2\omega^4}{k'G_pE_p} - \frac{\rho_pA_p\omega^2}{E_pI_p} - \left(\frac{\rho_p\omega^2}{k'A_pG_pE_p} - \frac{1}{E_pI_p}\right)(k_h+ic_h)$.

Substituting the displacement equation (27) into equations (39) and (40) are obtained:

$$\frac{d^4\bar{u}_p}{dz^4} + K\frac{d^2\bar{u}_p}{dz^2} + N\bar{u}_p = h_1A_st_1s^2e^{-sz} + h_2A_st_1e^{-sz} + h_3A_st_2e^{-\gamma z} \tag{41}$$

$$\frac{d^4\bar{\theta}_p}{dz^4} + K\frac{d^2\bar{\theta}_p}{dz^2} + N\bar{\theta}_p = h_4A_st_1se^{-sz} + h_4A_st_2\gamma e^{-\gamma z} \tag{42}$$

where, $h_1 = -\frac{(k_h+ic_h)}{k'A_pG_p}\cos\theta$, $h_2 = (k_h+ic_h)\left(\frac{1}{E_pI_p} - \frac{\rho_p\omega^2}{k'A_pG_pE_p}\right)\cos\theta$, $h_3 = h_2 + h_1\gamma^2$, $h_4 = \frac{(k'G_p+P)A_p}{E_pI_p}h_1$.

The ground solution of the homogeneous equation corresponding to the non-homogeneous linear ordinary differential equation (41) and equation (42) are:

$$u_p''(z) = M_a\cos(\lambda_az) + M_b\sin(\lambda_az) + M_dch(\lambda_bz) + M_esh(\lambda_bz) \tag{43}$$

$$\theta_p''(z) = M_f\cos(\lambda_az) + M_g\sin(\lambda_az) + M_hch(\lambda_bz) + M_ish(\lambda_bz) \tag{44}$$

where, $\lambda_a = \sqrt{\frac{K+\sqrt{K^2-4N}}{2}}$, $\lambda_b = \sqrt{\frac{-K+\sqrt{K^2-4N}}{2}}$.

The special solutions of equations (41) and (42) can be derived as:

$$\bar{u}_p^*(z) = A_sN_pe^{-sz} + A_sN_se^{-\gamma z} \tag{45}$$

$$\bar{\theta}_p^*(z) = A_sL_pe^{-sz} + A_sL_se^{-\gamma z} \tag{46}$$

in which: $N_p = \frac{(h_1s^2+h_2)t_1}{s^4+Ks^2+N}$, $N_s = \frac{h_3t_2}{\gamma^4+K\gamma^2+N}$ $L_p = \frac{h_4st_1}{s^4+Ks^2+N}$, $L_s = \frac{h_4\gamma t_2}{\gamma^4+K\gamma^2+N}$

The horizontal displacement amplitude and rotation angle of the pile can be expressed as:

$$\bar{u}_p = u_p''(z) + \bar{u}_p^*(z) \tag{47}$$

$$\bar{\theta}_p = \theta_p''(z) + \bar{\theta}_p^*(z) \tag{48}$$

Substituting equations (43) and (44) into equations (5) and (6) yield:

$$M_f = \chi_b M_b, M_g = \chi_a M_a, M_h = \chi_e M_e, M_i = \chi_d M_d \tag{49}$$

where, $\chi_a = -\chi_b = -\frac{k' A_p G_p \lambda_a + A_p P \lambda_a}{k' A_p G_p + \rho_p I_p \omega^2 + E_p I_p \lambda_a^2}$, $\chi_d = \chi_e = \frac{k' A_p G_p \lambda_b + A_p P \lambda_b}{k' A_p G_p + \rho_p I_p \omega^2 - E_p I_p \lambda_b^2}$,

The amplitude of the horizontal displacement and rotation angle of the pile can be expressed as:

$$\bar{u}_p = M_a \cos(\lambda_a z/H) + M_b \sin(\lambda_a z/H) + M_d ch(\lambda_b z/H) + M_e sh(\lambda_b z/H)$$
$$+ A_s N_p e^{-sz/H} + A_s N_s e^{-\gamma z/H} \tag{50}$$

$$\bar{\theta}_p = M_a \chi_a \sin(\lambda_a z/H) + M_b \chi_b \cos(\lambda_a z/H) + M_d \chi_d sh(\lambda_b z/H)$$
$$+ M_e \chi_e ch(\lambda_b z/H) + A_s L_p e^{-sz/H} + A_s L_s e^{-\gamma z/H} \tag{51}$$

According to elasticity, the bending moment and shear force along the pile shaft can be derived as:

$$\bar{M}_p = E_p I_p \left[ \begin{array}{l} M_a \chi_a \lambda_a \cos(\lambda_a z/H) - M_b \chi_b \lambda_a \sin(\lambda_a z/H) + M_d \chi_d \lambda_b ch(\lambda_b z/H) \\ + M_e \chi_e \lambda_b sh(\lambda_b z/H) - A_s s L_p e^{-sz/H} - A_s \gamma L_s e^{-\gamma z/H} \end{array} \right] \tag{52}$$

$$\bar{Q}_p = k' A_p G_p \left[ \begin{array}{l} -M_a (\lambda_a + \chi_a) \sin(\lambda_a z/H) + M_b (\lambda_a - \chi_b) \cos(\lambda_a z/H) + M_d (\lambda_b - \chi_d) sh(\lambda_b z/H) \\ + M_e (\lambda_b - \chi_e) ch(\lambda_b z/H) - A_s (s N_p + L_p) e^{-sz/H} - A_s (\gamma N_s + L_s) e^{-\gamma z/H} \end{array} \right] \tag{53}$$

Due to the pile head is in a flexible constraint state and the pile tip is in a fixed constraint state, the boundary conditions of the piles are:

$$\left\{ \begin{array}{l} \bar{M}_p(z) = k_0 \bar{\theta}_c + k_1 \left[ \bar{\theta}_p(0) - \bar{\theta}_c \right], z = 0 \\ \bar{Q}_p(z) - A_p P \bar{\theta}_p(z) + \omega^2 M \bar{u}_p(z) = 0, z = 0 \\ \bar{u}_p(z) = 0, z = H \\ \bar{\theta}_p(z) = 0, z = H \end{array} \right. \tag{54}$$

The boundary conditions of horizontal displacement, rotation angle, amplitude along the bending moment and shear force of the pile shaft can be solved:

$$M_a \cos(\lambda_a) + M_b \sin(\lambda_a) + M_d ch(\lambda_b) + M_e sh(\lambda_b)$$
$$+ A_s (N_p e^{-s} + N_s e^{-\gamma}) = 0 \tag{55}$$

$$M_a \chi_a \sin(\lambda_a) + M_b \chi_b \cos(\lambda_a) + M_d \chi_d sh(\lambda_b)$$
$$+ M_e \chi_e ch(\lambda_b) + A_s (L_p e^{-s} + L_s e^{-\gamma}) = 0 \tag{56}$$

$$M_a E_p I_p \chi_a \lambda_a - M_b k_1 \chi_b + M_d E_p I_p \chi_d \lambda_b - M_e k_1 \chi_e$$
$$- A_s (E_p I_p s L_p + E_p I_p \gamma L_s + k_1 L_p + k_1 L_s) = k_0 \bar{\theta}_c - k_1 \bar{\theta}_c \tag{57}$$

$$M_a (\omega^2 M) + M_b [k' A_p G_p (\lambda_a - \chi_b) - \chi_b A_p P] + M_d (\omega^2 M) + M_e [k' A_p G_p (\lambda_b - \chi_e) - \chi_e A_p P]$$
$$+ A_s [\omega^2 M (N_p + N_s) - A_p P (L_p + L_s) - k' A_p G_p (s N_p + L_p + \gamma N_s + L_s)] = 0 \tag{58}$$

From the continuity conditions of pile-soil interface displacement $u_r = u_p \cos \theta$, it can be obtained:

$$M_a + M_d + A_s \left( N_p + N_s + ik_R e^{-ik_R r} \right) = 0 \tag{59}$$

expressed in matrix form as:

$$\begin{bmatrix} \cos(\lambda_a) & \sin(\lambda_a) & ch(\lambda_b) & sh(\lambda_b) & N_p e^{-s}+N_s e^{-\gamma} \\ \chi_a \sin(\lambda_a) & \chi_b \cos(\lambda_a) & \chi_d sh(\lambda_b) & \chi_e ch(\lambda_b) & L_p e^{-s}+L_s e^{-\gamma} \\ E_p I_p \chi_a \lambda_a & -k_1 \chi_b & E_p I_p \chi_d \lambda_b & -k_1 \chi_e & -(E_p I_p s L_p + E_p I_p \gamma L_s + k_1 L_p + k_1 L_s) \\ \omega^2 M & k' A_p G_p (\lambda_a - \chi_b) & E_p I_p \chi_d \lambda_b & \omega^2 M & \omega^2 M_b (N_p+N_s) \\ & -\chi_b A_p P & & k' A_p G_p (\lambda_b & -A_p P (L_p+L_s) - k' A_p G_p \\ & & & -\chi_e - \chi_e A_p P & (s N_p + L_p + \gamma N_s + L_s) \\ 1 & 0 & 1 & 0 & N_p + N_s + ik_R e^{-ik_R r} \end{bmatrix} \begin{bmatrix} M_a \\ M_b \\ M_d \\ M_e \\ A_s \end{bmatrix} = \begin{bmatrix} 0 \\ 0 \\ k_0 \bar{\theta}_c - k_1 \bar{\theta}_c \\ 0 \\ 0 \end{bmatrix} \tag{60}$$

From the above process, all the undetermined parameters $M_a$, $M_b$, $M_d$, $M_e$, $A_s$ can be obtained, so as to give the analytical solutions of the horizontal displacement, rotation angle, bending moment and shear force along the pile shaft.

## 5 Numerical results and discussion

In this section, the proposed solution was validated through numerical examples, and parameters such as vertical load and flexible constraint were analyzed for pile displacement, rotation angle, bending moment, and shear force. The value was taken from reference [30]. Unless otherwise specified, the material properties used in this study are $\rho_s = 2.7 \times 10^3$ kg/m³, $E_a = 1 \times 10^{10}$ pa, $v = 0.35$; $d = 1$m, $r_0 = 0.5$m, $H = 20$m, $\rho_p = 2.5 \times 10^3$ kg/m³, $k' = 0.75$, $E_p = 2.5 \times 10^{10}$ Pa, $I_p = \frac{\pi}{32} m^4$, $v_p = 0.2$, $A_p = 0.25\pi m^2$.

### 5.1 Validation

To verify the correctness of the calculation model, the solutions of Makris [39] was compared to the model results. Set the boundary conditions of the calculation method given by Makris [39] as flexible constraints, and use the same parameters to validate the calculation model in this paper. Fig 2 compares the horizontal displacement and rotation with the results of Makri's solution [39]. The agreement between the two solutions is very high.

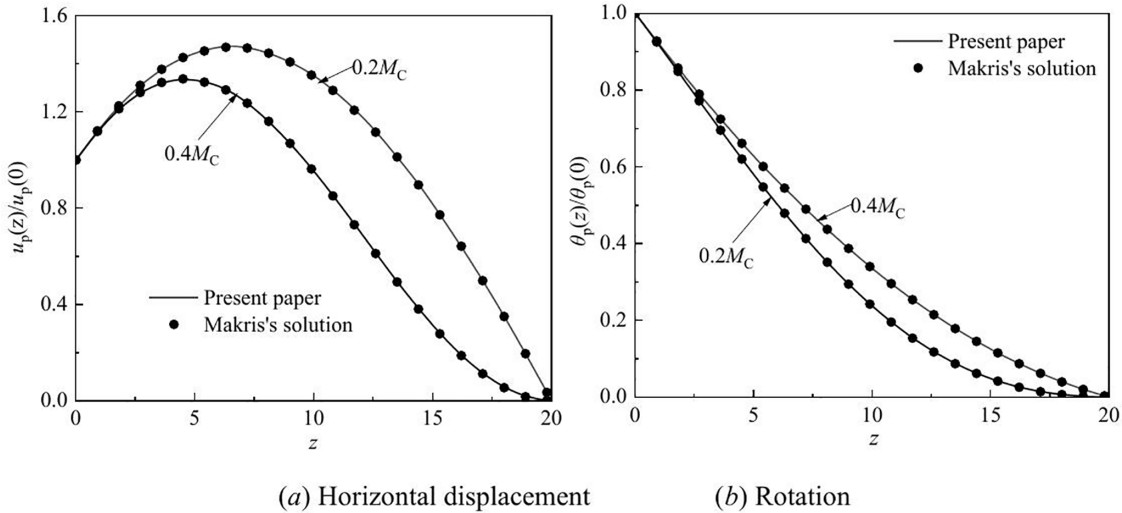

(a) Horizontal displacement  (b) Rotation

**Fig 2. Comparison of present solution with Makri's solution.**

## 5.2 Parameter analysis

The effect of vertical load on the lateral response of monopile under Rayleigh wave as a function of depth and frequency are shown in Fig 3 – Fig 8 for various values of the vertical load, flexible constraints, dimensionless frequency and Poisson's ratio. When the horizontal coordinate represents the depth "$z$", set the frequency $a_{effla}$ to 6. When the abscissa represents the frequency "$a_{effla}$", set the depth $z$ to 0.2. Some general trends are observed: as the depth increases, the displacement and shear force first increase and then decrease, while the rotation angle and bending moment gradually decrease. As the frequency increases, the displacement, rotation angle, bending moment, and shear force remain stable after resonance occurs in the resonance region.

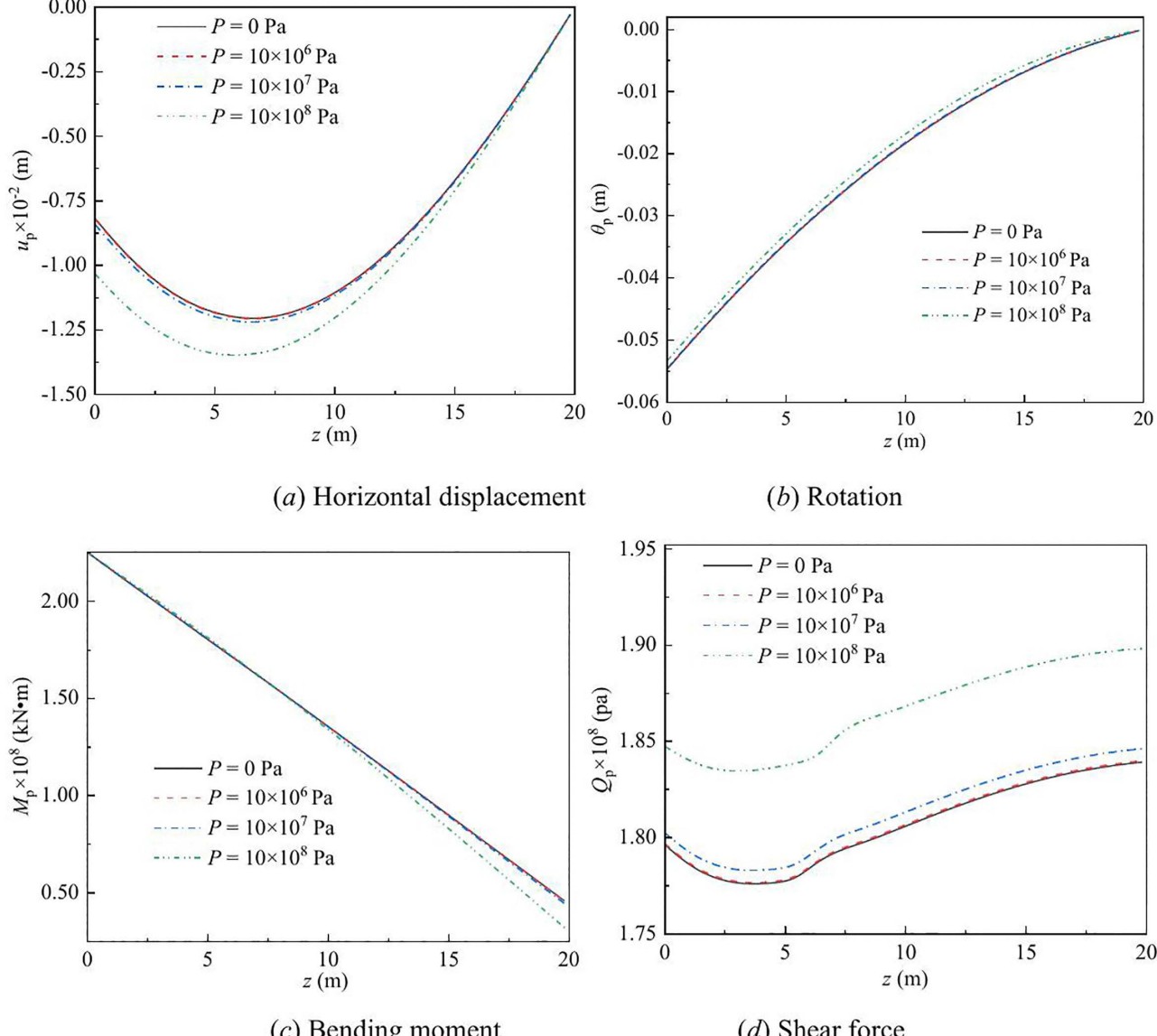

(a) Horizontal displacement

(b) Rotation

(c) Bending moment

(d) Shear force

**Fig 3. Effect of vertical load on the lateral response of monopile with depth.**

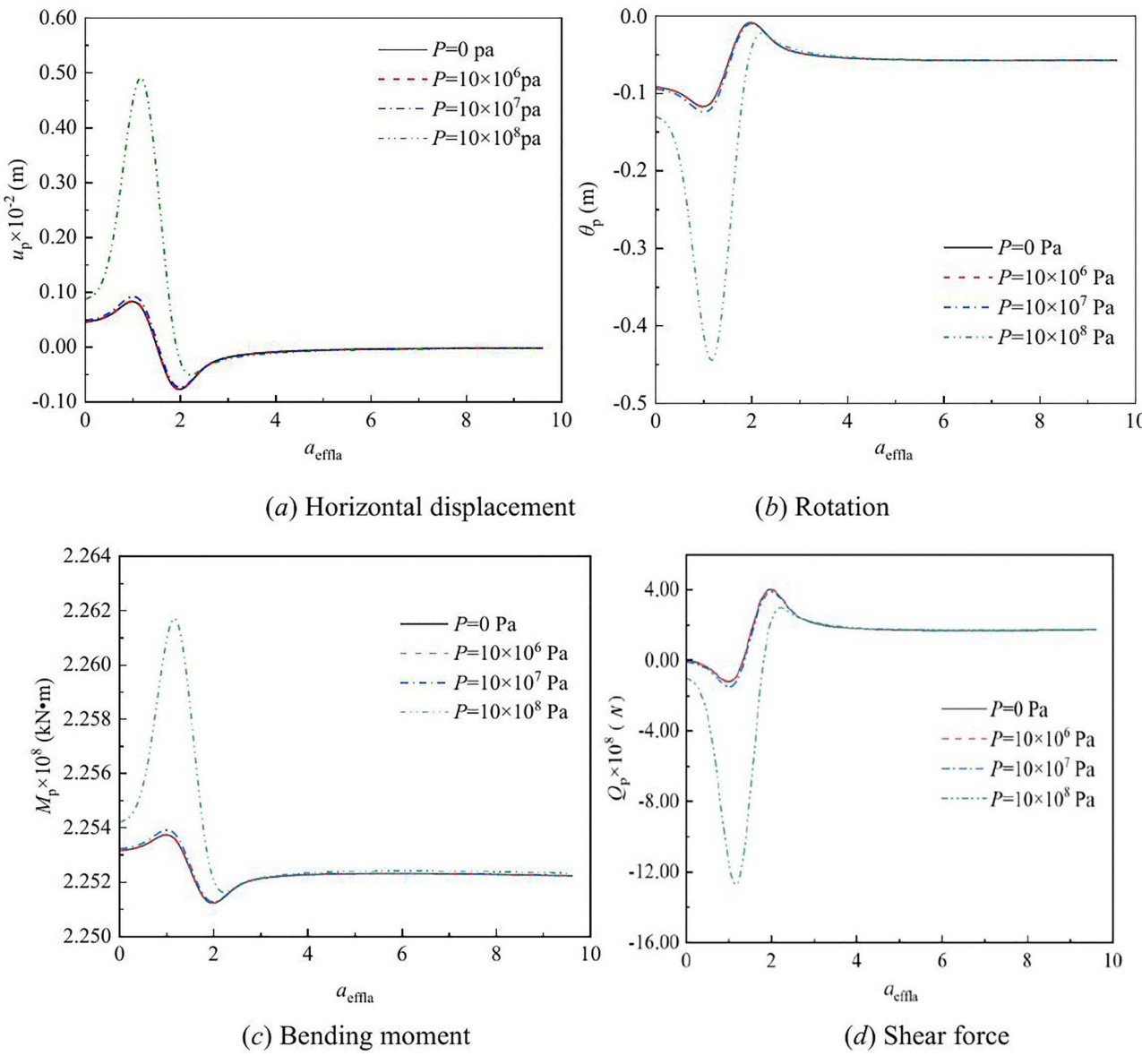

(a) Horizontal displacement

(b) Rotation

(c) Bending moment

(d) Shear force

**Fig 4. Effect of vertical load on the lateral response of monopile with frequency.**

For different values of vertical load, the variation in the displacement, rotation angle, bending moment, and shear force of the dynamic response are given in Fig 3 – Fig 4. From Fig 3, it can be observed that as the vertical load increases, the displacement and shear force increase, while the rotation angle and bending moment decrease. This is because, as the vertical load increases, the P-Δ effect becomes more pronounced, leading to increased horizontal displacement and shear force, while the rotation angle and bending moment decrease relatively.

Fig 4 shows the effect of vertical load on the displacement, rotation angle, bending moment, and shear force of monopile head with frequency. When the frequency is equal to the natural frequency, resonance occurs. In the resonance region, as the vertical load increases, the displacement of the pile head, corner bending moment, and shear force also increase. With the increase of frequency, the displacement of the pile head, bending moment, rotation Angle and shear force tend to stabilize.

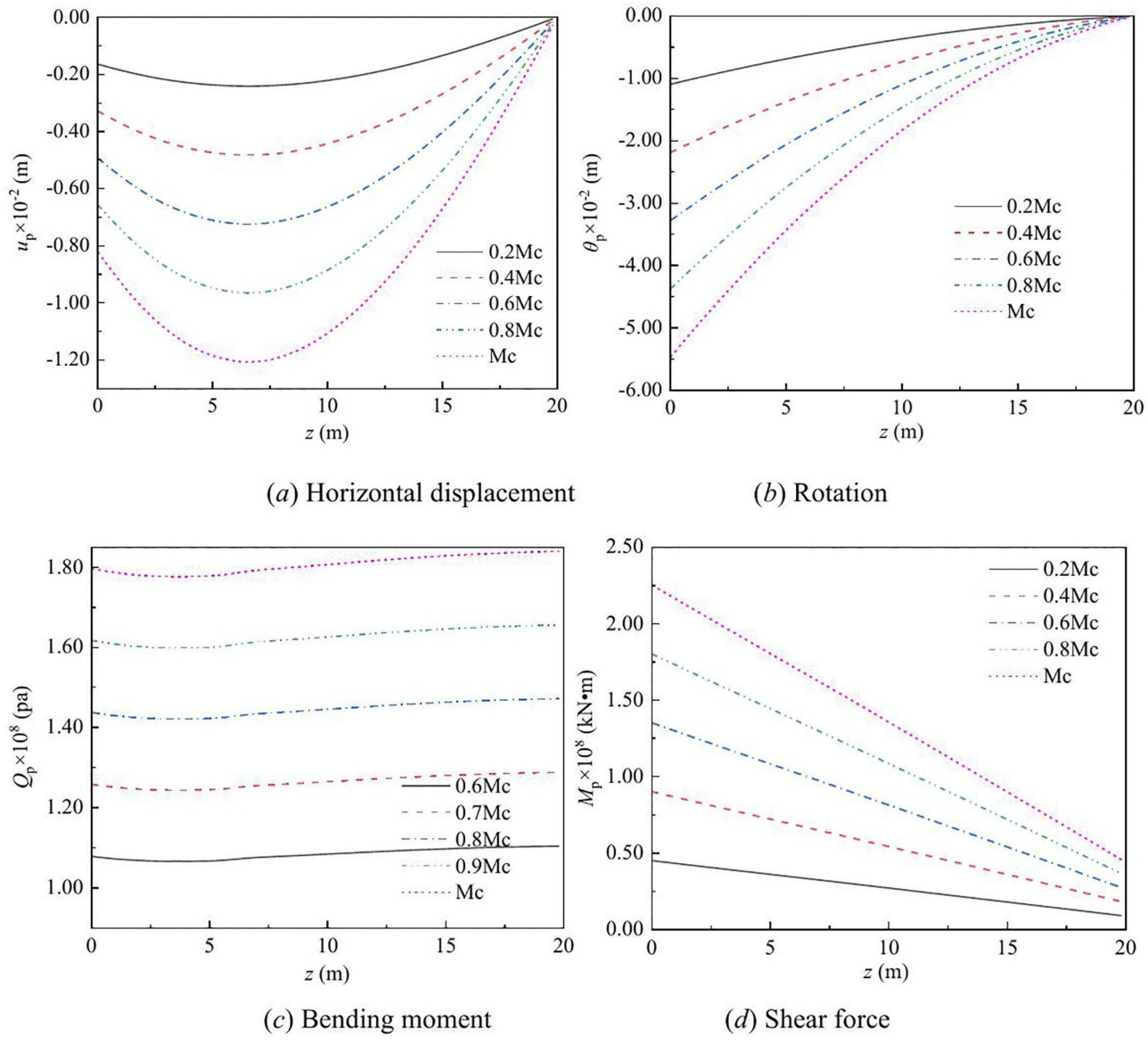

**Fig 5. Effect of flexible constraints on the lateral response of monopile with depth.**

For different values of flexible constraints, the variation in the displacement, rotation angle, bending moment, and shear force of the dynamic response are given in Fig 5 – Fig 6. From Fig 5, it can be observed that as the flexible constraints increases, the displacement, rotation angle, bending moment, and shear force of the pile all increase. This is because after the flexible constraint is enhanced, the rotation of the pile head is restricted, the coordinated deformation of the pile and soil is strengthened, and more horizontal seismic forces are transmitted to the pile body through the soil, resulting in the synchronous amplification of the displacement, rotation Angle, bending moment and shear force of the pile.

From Fig 6, it can be observed that the displacement, rotation angle, bending moment, and shear force of the pile increase with the increase of flexible constraints before resonance. The same variation pattern is observed in the resonance region. After resonance, as the frequency increases, the displacement tends to stabilize, and the rotation angle,

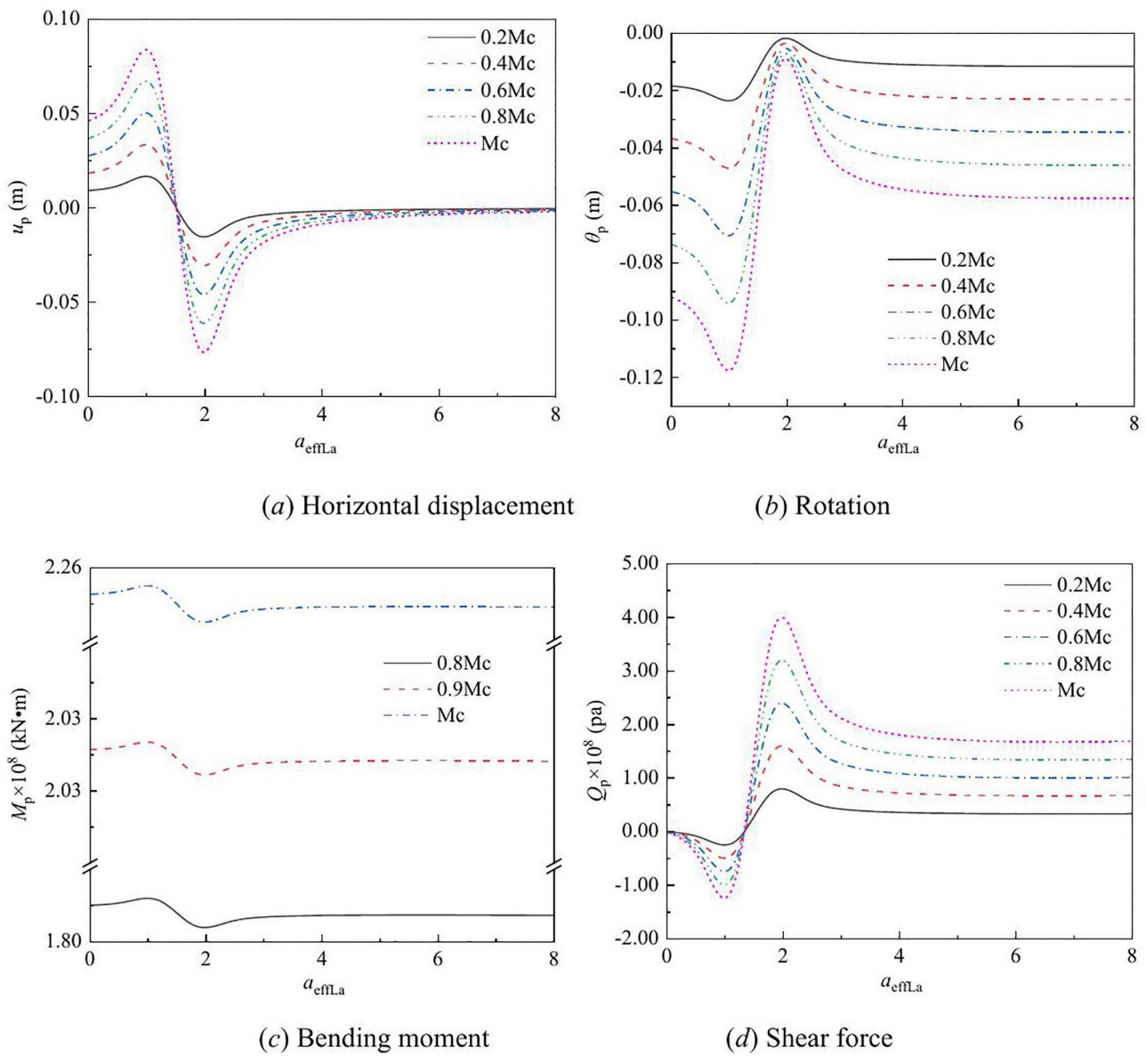

**Fig 6. Effect of flexible constraints on the lateral response of monopile with frequency.**

bending moment, and shear force increase with the increase of flexible constraints. This is because after the flexible constraint is enhanced, the rotation of the pile head is more restricted, the overall stiffness of the system increases, and the energy during resonance is more likely to be concentrated on the pile body, resulting in a simultaneous increase in the displacement, rotation Angle, bending moment and shear force at the resonance zone. After entering the high-frequency band, the inertial effect takes the lead, and the displacement tends to stabilize. However, as the constraint stiffness is still increasing, the rotation Angle, bending moment and shear force continue to increase with the increase of flexible constraints.

For different values of dimensionless frequency, the variation in the displacement, rotation angle, bending moment, and shear force of the dynamic response is given in Fig 7. When the dimensionless frequency changes from 4 to 10, as the

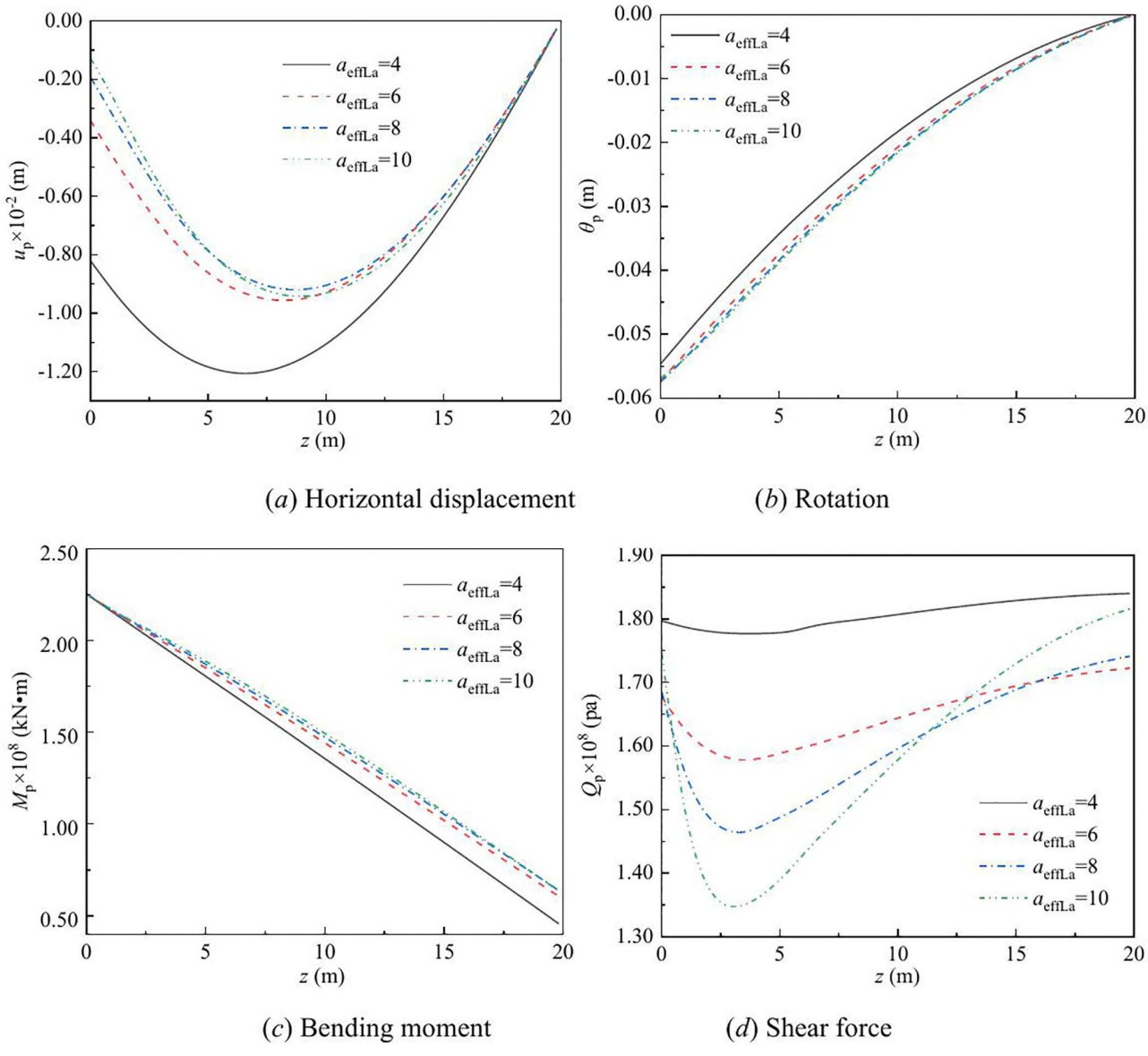

(a) Horizontal displacement (b) Rotation

(c) Bending moment (d) Shear force

**Fig 7. Effect of dimensionless frequency on the lateral response of monopile with depth.**

frequency increases, the displacement and shear force decrease, while the rotation angle and bending moment increase. This is because under high-frequency excitation, the inertial stiffness of the soil increases, the displacement and shear force are suppressed, and the rotation Angle and bending moment increase concentratedly accordingly.

For different values of Poisson's ratio, the variation in the displacement, rotation angle, bending moment, and shear force of the dynamic response is given in Fig 8. When the Poisson's ratio of soil changes from 0.25 to 0.35, as the frequency increases, the displacement and shear force increase, while the rotation angle and bending moment decrease. This is because as the Poisson's ratio increases, the lateral swelling of the soil intensifies, the horizontal stiffness decreases, and the displacement and shear force increase accordingly, while the rotation Angle and bending moment relatively decrease.

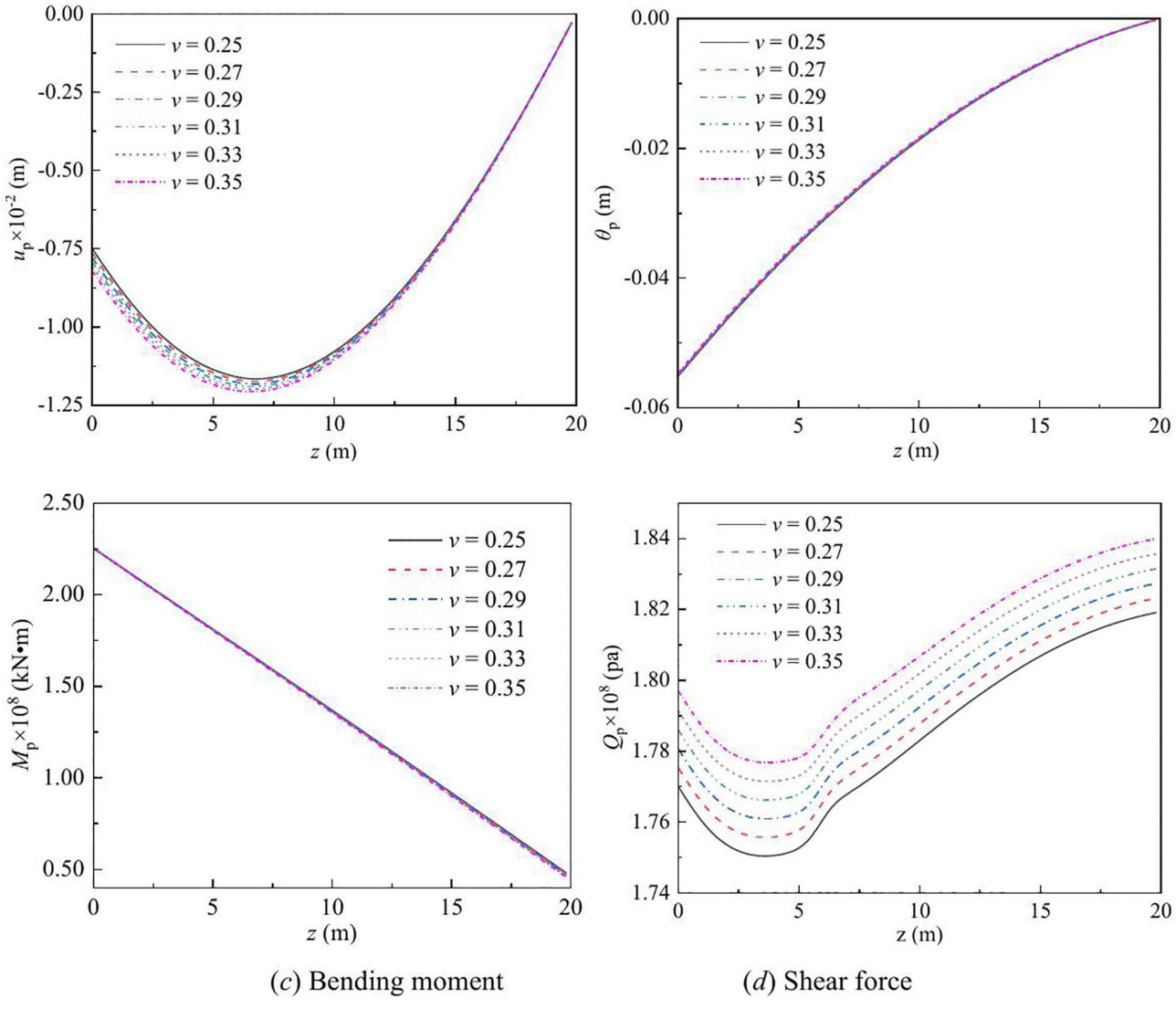

**Fig 8. Effect of Poisson's ratio on the lateral response of monopile with depth.**

## 6 Conclusion

In this paper, considering the effect of vertical load on the lateral response of monopile under Rayleigh wave, a computational model for the lateral response of monopile under Rayleigh wave was established. By setting a series of numerical results, the lateral response of monopile was studied. Based on this study, the following conclusions are obtained:

(1) As the depth increases, the displacement and shear force first increase and then decrease, while the rotation angle and bending moment gradually decrease.

(2) As the frequency increases, the displacement, rotation angle, bending moment, and shear force remain stable after resonance occurs in the resonance region.

(3) As the vertical load Poisson's and increases, the displacement and shear force increase, while the rotation angle and bending moment decrease.

(4) As the flexible constraints increases, the displacement, rotation angle, bending moment, and shear force of the pile all increase.

## Supporting information

**S1 Data. The data in Figs 2–8.**
(XLSX)

## Author contributions

**Data curation:** Sitong Zhao.

**Formal analysis:** Xiaohui Zhang.

**Funding acquisition:** Zhiqiang Wang.

**Investigation:** Ruming Ma, Fanbao Meng, Zhiqiang Wang.

**Methodology:** Xiaohui Zhang.

**Project administration:** Ruming Ma, Fanbao Meng.

**Software:** Hui Yan.

**Validation:** Hui Yan.

**Writing – original draft:** Sitong Zhao.

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
