## [Decision Letter · Decision Letter 0]

14 Aug 2025

PONE-D-25-40516Analysis of horizontal dynamic response of single pile under vertical load-Rayleigh wave combined actionPLOS ONE

Dear Dr. Zhao,

Thank you for submitting your manuscript to PLOS ONE. After careful consideration, we feel that it has merit but does not fully meet PLOS ONE’s publication criteria as it currently stands. Therefore, we invite you to submit a revised version of the manuscript that addresses the points raised during the review process.

We look forward to receiving your revised manuscript.

Kind regards,

Zhipeng Zhao

Academic Editor

PLOS ONE

Journal Requirements:

“This work was supported by the National Natural Science Foundation of China (.52408368). Lanzhou University of Technology Hongliu outstanding young talent support program, Key R&D Plan of Gansu Province(23YFFA0063). 2024 University teacher Innovation Fund project (2024A-019). Gansu Provincial Key R&D Program (25YFGE002). Tianshui Science and Technology Support Program (TS-STK-2024A-283).”

4. Thank you for stating the following in your Competing Interests section: “no”

5. We note that your Data Availability Statement is currently as follows: All relevant data are within the manuscript and its Supporting Information files.

7. PLOS requires an ORCID iD for the corresponding author in Editorial Manager on papers submitted after December 6th, 2016. Please ensure that you have an ORCID iD and that it is validated in Editorial Manager. To do this, go to ‘Update my Information’ (in the upper left-hand corner of the main menu), and click on the Fetch/Validate link next to the ORCID field. This will take you to the ORCID site and allow you to create a new iD or authenticate a pre-existing iD in Editorial Manager.

Reviewers' comments:

Reviewer's Responses to Questions

**Comments to the Author**

1. Is the manuscript technically sound, and do the data support the conclusions?

Reviewer #1: Yes

Reviewer #2: Yes

Reviewer #3: Yes

2. Has the statistical analysis been performed appropriately and rigorously? 

Reviewer #1: Yes

Reviewer #2: No

Reviewer #3: Yes

3. Have the authors made all data underlying the findings in their manuscript fully available?

Reviewer #1: No

Reviewer #2: No

Reviewer #3: Yes

4. Is the manuscript presented in an intelligible fashion and written in standard English?

Reviewer #1: Yes

Reviewer #2: Yes

Reviewer #3: Yes

5. Review Comments to the Author

Reviewer #1: The study is novel and publishable, I recommend its publication after correcting the followings:

1- For the soil (foundation) structure interactions and different foundation models consider the suggested refs below to summarize the state of the art in the introduction section.

2- Cite your formulations.

3- For the dynamic interactions of the system, discuss relationship the wave frequency and the eigen frequencies of the system.

For the dynamic interaction and the foundation models see:

https://doi.org/10.1016/j.ijmecsci.2019.01.033.

https://doi.org/10.1080/15397734.2021.1999263.

https://doi.org/10.1080/15397734.2021.1904255

Reviewer #2: This research focuses on the key issues of pile-soil interaction under the action of Rayleigh wave. The influence of vertical loads on the horizontal dynamic response of piles under Rayleigh waves were studied by the Timoshenko beam theory and elastic dynamic equation. By setting a series of numerical results, the obtained conclusions provide a practical technical method for the dynamic design of pile foundations. However, the innovation of this paper is lacking and should be rejected.

(1)The paper failed to elaborate on the innovative aspects of the presented model in its introduction. In fact, this paper is very similar to the calculation model proposed by Makris [39] and Zhang et al. [31] and other scholars.

(2)In the parameter analysis of the paper, only a preliminary description of the curve's changing pattern was provided, without considering the underlying mechanism.

(3)A summary of the limitations and innovations of the model should be provided to facilitate readers' understanding of the entire paper.

(4) Under Rayleigh seismic waves, is the deformation of the pile-soil system elastic? The paper seems to have failed to discuss the application or assumption conditions of the proposed model.

Reviewer #3: Comments

This topic is interesting. An analytical model is presented to discuss the horizontal dynamic behavior of a single pile under vertical load-Rayleigh wave combined action. The analytical solutions of the horizontal displacement, rotation angle, bending moment and shear force along the pile shaft are derived in detail. A more in-depth discussion on behaviour of the results is also needed. The specific comments are given below.

(1) When mathematical symbols first appear, the authors should provide clear definitions to enhance readability. For instance, symbols such as θc and θp in Eq.(1), δij in Eq. (4), and kh, ch in Eq. (5). Similar issues occur throughout the manuscript. The authors are advised to thoroughly check the manuscript and make necessary corrections.

(2) Regarding Eq. (20), how was the complex wavenumber kR obtained? Please clarify its calculation method and theoretical basis.

(3) Variables should be typeset in italics to adhere to academic writing conventions. For example, the depth variable ‘z’ appears in upright (roman) font in multiple equations and figures throughout the manuscript. The authors should carefully review the entire manuscript and correct such issues.

(4) In Section 5 (Numerical results and discussion): The authors should provide justification or references for the parameter values used. Furthermore, units must be specified for parameters such as Ip and Ap.

(5) The authors mention that the abscissa (x-axis) in Figure 4 is related to frequency. This relationship should be explicitly stated in an appropriate location. Additionally, the unit for the ordinate (y-axis) in the shear force graph is incorrectly written. The authors are requested to verify the correctness of all units throughout the manuscript.

(6) In Figure 6c: why are there fewer analysis cases compared to the other three subfigures (6a, 6b, 6d)? Why do two ‘2.03’ appear on the vertical coordinate? Please check and correct.

(7) The authors may find some inspirations in the following reference. which can enrich the literature review part of this manuscript if cited.

Reliability and sensitivity analyses of monopile supported offshore wind turbines based on probability density evolution method with pre-screening of controlling parameters[J]. Ocean Engineering, 2024, 310: 118746.

A novel analytical solution for horizontal vibration of partially embedded offshore piles considering the distribution effect of wave loads [J]. Ocean Engineering, 2024, 307: 118179

Analytical solution for horizontal vibration of end-bearing single pile in radially heterogeneous saturated soil[J]. Applied Mathematical Modelling, 2023, 116:65–83. https://doi.org/10.1016/j.apm.2022.11.027

A new analytical solution for horizontal vibration of floating pile in saturated soil based on FSSP method [J]. Soil Dynamics and Earthquake Engineering, 2024, 187: 108960. https://doi.org/10.1016/j.soildyn.2024.108960

A close-formed solution for the horizontal vibration of a pipe pile in saturated soils considering the radial heterogeneity effect[J]. Computers and Geotechnics, 2023, 158: 105379. DOI:10.1016/j.compgeo.2023.105379

Analytical solution for longitudinal vibration of a floating pile in saturated porous media based on a fictitious saturated soil pile model [J]. Computers and Geotechnics.2021,131:103942.

(8) The analysis in Section 5.2 should be significantly enriched. Currently, the authors merely describe the trends of various responses with changing parameters without analyzing the underlying physical mechanisms responsible for these observed trends.

(9) In Section 5.2 (Parameter Analysis), it is stated: ‘The effect of vertical load on the lateral response of monopile under Rayleigh wave as a function of depth and frequency are shown in Figure 3 - Figure 8 for various values of the vertical load, flexible constraints, dimensionless frequency and Poisson's ratio.’ When the abscissa (horizontal axis) represents depth ‘z’, should the frequency be a fixed value? Conversely, when the abscissa represents aeffla, should the depth ‘z’ also be a fixed value for the same data points? The manuscript does not clarify how these parameters were controlled in the analysis. The authors should explain this explicitly.

6. PLOS authors have the option to publish the peer review history of their article (what does this mean? ). If published, this will include your full peer review and any attached files.

**Do you want your identity to be public for this peer review?** For information about this choice, including consent withdrawal, please see our Privacy Policy .

Reviewer #1: No

Reviewer #2: No

Reviewer #3: No

---

## [Author Response · Author response to Decision Letter 1]

8 Oct 2025

Reviewer 1:

1- For the soil (foundation) structure interactions and different foundation models consider the suggested refs below to summarize the state of the art in the introduction section.

Response Thank you very much for your valuable comments. We have summarized the soil-structure interaction in the introduction based on the references you provided. For details, please refer to References [20-22].

2- Cite your formulations.

Response Thank you very much for your valuable comments. We have already cited the latest formula in the basic equation.

3- For the dynamic interactions of the system, discuss relationship the wave frequency and the Eigen frequencies of the system.

For the dynamic interaction and the foundation models see:https://doi.org/10.1016/j.ijmecsci.2019.01.033. https://doi.org/10.1080/15397734.2021.1999263. https://doi.org/10.1080/15397734.2021.1904255

Response Based on the literature you provided and the latest research results, we have summarized the dynamic interaction system.

Reviewer 2:

(1)The paper failed to elaborate on the innovative aspects of the presented model in its introduction. In fact, this paper is very similar to the calculation model proposed by Makris [39] and Zhang et al. [31] and other scholars.

Response Makris [39] and Zhang et al. [31] and other scholars. The studies of Makris [39] and Zhang et al. [31] and other scholars have focused on the responses of soil and piles under Rayleigh wave action.

In actual engineering, pile foundations are subject to the combined action of vertical and horizontal dynamic loads. The existence of vertical loads will produce a second-order effect, leading to an increase in horizontal displacement. However, most of the existing studies have not taken into account the combined effect of vertical and horizontal loads. Considering the influence of vertical loads on the horizontal vibration characteristics of pile foundations in non-saturated soil still requires further in-depth research. This paper studies the influence of vertical loads and investigates the dynamic response of pile foundations under Rayleigh wave action.

(2)In the parameter analysis of the paper, only a preliminary description of the curve's changing pattern was provided, without considering the underlying mechanism.

Response Thank you very much for your valuable comments. Based on your opinion, we have added the mechanism that leads to the results in the parameter analysis process.

(3)A summary of the limitations and innovations of the model should be provided to facilitate readers' understanding of the entire paper.

Response Thank you very much for your valuable comments. Based on your opinion, we have already explained the applicability of this study in the last paragraph of the abstract and introduction.

(4) Under Rayleigh seismic waves, is the deformation of the pile-soil system elastic? The paper seems to have failed to discuss the application or assumption conditions of the proposed model.

Response�A review of a large number of research results reveals that under the action of Rayleigh waves, elastic deformation occurs in the pile-soil system. In actual engineering, the pile-soil systems of Rayleigh waves caused by mechanical and human activities are all elastic. Under seismic action, the pile-soil system in the far field is also elastic. This research achievement can be applied to the pile-soil interaction system of Rayleigh waves caused by mechanical and human activities as well as far-field Rayleigh waves generated by earthquakes.

Reviewer 3:

(1) When mathematical symbols first appear, the authors should provide clear definitions to enhance readability. For instance, symbols such as θc and θp in Eq.(1), δij in Eq. (4), and kh, ch in Eq. (5). Similar issues occur throughout the manuscript. The authors are advised to thoroughly check the manuscript and make necessary corrections.

Response Thank you very much for your valuable comments.Based on your opinion�we have already provided definitions and explanations of the relevant symbols below formulas (1), (4), and (5).

(2) Regarding Eq. (20), how was the complex wavenumber kR obtained? Please clarify its calculation method and theoretical basis.

Response Thank you very much for your valuable comments.Based on your opinion, We have provided relevant explanations of the complex wavenumber kR below formula (20) and given the theoretical basis.

(3) Variables should be typeset in italics to adhere to academic writing conventions. For example, the depth variable ‘z’ appears in upright (roman) font in multiple equations and figures throughout the manuscript. The authors should carefully review the entire manuscript and correct such issues.

Response Thank you very much for your valuable comments.Based on your opinion We have carefully reviewed the entire text and have already corrected the font of some parts that were not written properly, such as the depth variable "z".

(4) In Section 5 (Numerical results and discussion): The authors should provide justification or references for the parameter values used. Furthermore, units must be specified for parameters such as Ip and Ap.

Response Thank you very much for your valuable comments.Based on your opinion In Section 5 (Numerical Results and Discussion), we added references to the relevant data and specified the units for parameters such as Ip and Ap.

(5) The authors mention that the abscissa (x-axis) in Figure 4 is related to frequency. This relationship should be explicitly stated in an appropriate location. Additionally, the unit for the ordinate (y-axis) in the shear force graph is incorrectly written. The authors are requested to verify the correctness of all units throughout the manuscript.

Response Thank you very much for your valuable comments.Based on your opinion We have added the relationship of the corresponding parameters varying with frequency below Figure 4 and modified the unit of the vertical coordinate in the shear force diagram.

(6) In Figure 6c: why are there fewer analysis cases compared to the other three subfigures (6a, 6b, 6d)? Why do two ‘2.03’ appear on the vertical coordinate? Please check and correct.

Response Thank you very much for your valuable comments.Based on your opinion Compared with the high-constraint state, the bending moment value is very small under the low-constraint state, resulting in an unintuitive graph drawing. Moreover, in Figure 6c, we analyzed the changes of bending moments under three different high-constraint states, which can already prove the influence law of constraints on bending moments.

The appearance of two "2.03" in the vertical coordinate is due to the significant difference in their values, which makes the graph less intuitive. Therefore, some data is extracted for comparison.

(7) The authors may find some inspirations in the following reference. which can enrich the literature review part of this manuscript if cited.

Reliability and sensitivity analyses of monopile supported offshore wind turbines based on probability density evolution method with pre-screening of controlling parameters[J]. Ocean Engineering, 2024, 310: 118746.

A novel analytical solution for horizontal vibration of partially embedded offshore piles considering the distribution effect of wave loads [J]. Ocean Engineering, 2024, 307: 118179

Analytical solution for horizontal vibration of end-bearing single pile in radially heterogeneous saturated soil[J]. Applied Mathematical Modelling, 2023, 116:65–83. https://doi.org/10.1016/j.apm.2022.11.027

A new analytical solution for horizontal vibration of floating pile in saturated soil based on FSSP method [J]. Soil Dynamics and Earthquake Engineering, 2024, 187: 108960. https://doi.org/10.1016/j.soildyn.2024.108960

A close-formed solution for the horizontal vibration of a pipe pile in saturated soils considering the radial heterogeneity effect[J]. Computers and Geotechnics, 2023, 158: 105379. DOI:10.1016/j.compgeo.2023.105379

Analytical solution for longitudinal vibration of a floating pile in saturated porous media based on a fictitious saturated soil pile model [J]. Computers and Geotechnics.2021,131:103942.

Response Thank you very much for your valuable comments. Based on your opinion, We have revised some of the introductions based on the literature you provided. For details, please refer to References [3,14-15,34-35].

(8) The analysis in Section 5.2 should be significantly enriched. Currently, the authors merely describe the trends of various responses with changing parameters without analyzing the underlying physical mechanisms responsible for these observed trends.

Response Thank you very much for your valuable comments.Based on your opinion.In Section 5.2 of the Pattern Analysis, we have added the underlying physical mechanisms that lead to these observed trends

(9) In Section 5.2 (Parameter Analysis), it is stated: ‘The effect of vertical load on the lateral response of monopile under Rayleigh wave as a function of depth and frequency are shown in Figure 3 - Figure 8 for various values of the vertical load, flexible constraints, dimensionless frequency and Poisson's ratio.’ When the abscissa (horizontal axis) represents depth ‘z’, should the frequency be a fixed value? Conversely, when the abscissa represents aeffla, should the depth ‘z’ also be a fixed value for the same data points? The manuscript does not clarify how these parameters were controlled in the analysis. The authors should explain this explicitly.

Response Thank you very much for your valuable comments.Based on your opinion. In the 5.2 parameter analysis, we have explained the variable control of frequency and depth.

---

## [Decision Letter · Decision Letter 1]

30 Oct 2025

Analysis of horizontal dynamic response of single pile under vertical load-Rayleigh wave combined action

PONE-D-25-40516R1

Dear Dr. Zhao,

We’re pleased to inform you that your manuscript has been judged scientifically suitable for publication and will be formally accepted for publication once it meets all outstanding technical requirements.

Kind regards,

Zhipeng Zhao

Academic Editor

PLOS ONE

Additional Editor Comments (optional):

Reviewers' comments:

Reviewer's Responses to Questions

**Comments to the Author**

1. If the authors have adequately addressed your comments raised in a previous round of review and you feel that this manuscript is now acceptable for publication, you may indicate that here to bypass the “Comments to the Author” section, enter your conflict of interest statement in the “Confidential to Editor” section, and submit your "Accept" recommendation.

Reviewer #1: All comments have been addressed

Reviewer #3: (No Response)

2. Is the manuscript technically sound, and do the data support the conclusions?

Reviewer #1: Yes

Reviewer #3: Yes

3. Has the statistical analysis been performed appropriately and rigorously? 

Reviewer #1: N/A

Reviewer #3: Yes

4. Have the authors made all data underlying the findings in their manuscript fully available?

Reviewer #1: Yes

Reviewer #3: Yes

5. Is the manuscript presented in an intelligible fashion and written in standard English?

Reviewer #1: Yes

Reviewer #3: Yes

6. Review Comments to the Author

Reviewer #1: Accept. The Authors have adressed all comments, thus I strongly recommended publishing of the study.

Reviewer #3: The authors have addressed all the points of the reviewers' comments. I recommend that this manuscript can be accepted for further publication

7. PLOS authors have the option to publish the peer review history of their article (what does this mean? ). If published, this will include your full peer review and any attached files.

**Do you want your identity to be public for this peer review?** For information about this choice, including consent withdrawal, please see our Privacy Policy .

Reviewer #1: No

Reviewer #3: No

---

## [Editor Report · Acceptance letter]

PONE-D-25-40516R1

PLOS ONE

Dear Dr. Zhao,

I'm pleased to inform you that your manuscript has been deemed suitable for publication in PLOS ONE. Congratulations! Your manuscript is now being handed over to our production team.

Kind regards,

on behalf of

Dr. Zhipeng Zhao

Academic Editor

PLOS ONE